# The Effect of Strength Training on Undetected Shoulder Pathology in Asymptomatic Athletes: An MRI Observational Study

**DOI:** 10.3390/sports10120210

**Published:** 2022-12-15

**Authors:** Emil Noschajew, Alexander Azesberger, Felix Rittenschober, Amadeus Windischbauer, Michael Stephan Gruber, Reinhold Ortmaier

**Affiliations:** Department of Orthopedic Surgery, Ordensklinikum Linz Barmherzige Schwestern, Vinzenzgruppe Center of Orthopedic Excellence, Teaching Hospital of the Paracelsus Medical University, 5020 Salzburg, Austria

**Keywords:** athletes, bodybuilding, MRI, rotator cuff, shoulder, resistance training

## Abstract

Background: Data on the effects of weight training on joint morphology are mostly restricted to muscle gain. However, in many circumstances, it is not stated if there are negative consequences for the joints and their surrounding components. This study was conducted to explore whether long-term excessive resistance training (RT) causes hidden pathological alterations in the shoulder. Methods: A total of eleven asymptomatic sportsmen (22 shoulders) underwent clinical and radiological examination of both shoulder joints. All participants had engaged in bodybuilding for at least four years, at least three times per week, and for at least four hours per week. All participants were examined clinically using the Constant Murley Score (CMS), Simple Shoulder Test (SST), UCLA Activity Test, and a specially designed questionnaire. All participants received a bilateral shoulder MRI. The MRI scans were checked for pathology using a checklist. Results: Maximum scores were observed for the SST and UCLA Activity Test. The CMS was 97.7 points on average (range, 87–100). RT had been conducted for a mean of 10.7 years (range, 4–20), for an average of 8.8 h a week (range, 4–12). MRI examinations revealed two supraspinatus tendinopathies (9.1%), one labral change (4.5%), three humeral tuberosity cysts (13.6%), fourteen acromioclavicular (AC) joint hypertrophies (63.6%), five AC joint osteophytes (22.7%), and ten signs of AC joint inflammation (45.5%). Conclusions: The research results show that strength is associated with MRI-documented AC joint pathology. However, it appears that RT may not negatively affect other anatomical structures of the shoulder.

## 1. Introduction

The shoulder has a unique anatomical structure that may make it prone to both acute and chronic injuries [1]. In orthopedic and trauma surgery of the shoulder, not only degenerative disorders such as osteoarthritis but also sports-related injuries play a significant role [2]. Repetitive motion has been shown to affect the shoulder and can lead to changes in sports primarily characterized by throwing and overhead movements of the arm [2]. These changes are often asymptomatic at first and may only lead to difficulties later on [2]. RT is becoming more popular. For the purposes of injury prevention, rehabilitation, and fitness-related activities, RT has been recommended as a way to build muscle performance. RT has been shown to have many advantages, but there is also a potential for harm. Both competitive and leisure RT has been linked to injuries [3]. RT injuries span from acute to adaptive complications, with improper use of the equipment and losing control when using free weights being known causes of trauma [3,4]. However, it is unknown whether continuous RT performed several times a week for years leads to hidden changes to the glenohumeral joint and surrounding structures [3,5]. There are only a few studies on this topic in the literature. The purpose of this study was to determine if persistent, long-term RT of the shoulder results in undetected pathological changes. With this study, we can initiate further studies about hidden morphological changes in the shoulder and AC joint after RT. Furthermore, it could help us draw conclusions about the possible cause of pathological changes and idiopathic injuries in the shoulder and AC joint. This study contributes to answering the question of whether RT negatively affects shoulder morphology and whether it is associated with hidden asymptomatic shoulder pathologies [2,3,5,6].

## 2. Materials and Methods

This study was conducted as a prospective descriptive study. Eleven subjects (22 shoulders) were recruited from October 2019 to August 2020. All subjects had been bodybuilding athletes for at least four years. All of them undertook at least three bodybuilding sessions per week for at least four hours a week. Exclusion criteria were previous shoulder surgeries or known shoulder pathologies. Both shoulders of all participants were examined, resulting in a caseload of 22 asymptomatic shoulders. All participants were male, with a mean age of 32.5 years (range, 24–48; SD, 7.3), a mean height of 177.8 cm (range, 170–186; SD, 4.3), and a mean weight of 89.4 kg (range, 74–104; SD, 10.1). The participants were evaluated clinically and radiographically. All of them had at least four years of training prior to the examination. Seven participants, most of whom were older than the mean age of 32.5 years, had been training for more than nine years prior to the examination. A questionnaire was used to query and document height and weight for BMI determination (see Table 1), the duration of their RT career, average training duration and frequency per week, previous injuries, and the number of previous participations in professional bodybuilding competitions. The second part of the questionnaire consisted of five scales with zero to ten points for shoulder pain during and after sports, pain in other joints, uncertainty and fear of injury and dislocation during sports, and feeling of instability during sports. Furthermore, the CMS, SST, and UCLA Activity Test were used for clinical assessment [7,8,9].

The MRI examination was performed using a Siemens Skyra with a field strength of three Tesla. The review of the MRI images was performed by a radiologist and a research associate using a checklist (see Table 2) of pathologies according to the method of Barreto et al. [10]. In the event of a disagreement between the two reviewers’ findings, a third radiologist was consulted to reach a decision.

Descriptive statistics were used for data evaluation. The data were evaluated and compared using SPSS^®^ 26.0 software (IBM Corp, Armonk, NY, USA). All methods were used to comply with ethical standards and the requirements of the ethical review board of the Ordensklinikum Barmherzige Schwestern and Barmherzige Brüder Linz. This study was approved by the joint Ethics Committee of the Ordensklinikum Barmherzigen Schwestern and Barmherzige Brüder Linz (study number EKS 22/19).

## 3. Results

### 3.1. Clinical Outcome

On average, the eleven participants had been bodybuilding for 10.7 years (range, 4 to 20) and were training 4.7 times per week (range, 3 to 7) for an average of 8.8 h (range, 4 to 12) per week. The questionnaire revealed that most participants had little to no pain in the categories assessed (Table 3).

On average, the CMS was 98.1 points (range, 89 to 100) for the right shoulder and 97.3 points (range, 87 to 100) for the left shoulder. CMS results for all 22 shoulders showed a mean score of 97.7 points (range, 87 to 100). Of these 22 shoulders, 19 (86.4%) had no limitations or pain, and 3 (13.6%) had mild discomfort that, upon further inquiry, was due to recent previous exercise and heavy loading. In terms of mobility, all subjects achieved the maximum range of motion (Table 4).

Regarding their job, working height, and sleep quality, all participants stated that they did not experience any limitations or pain in their shoulders. Only 2 of the 22 shoulders (9.1%) had a minor limitation in terms of leisure activities. All participants could perform the range of motion and strength measurements without restrictions. In the SST, all subjects achieved the maximum score for both arms in this test. The same applies to the UCLA Activity Test.

### 3.2. Radiological Outcome

Thirty-five abnormalities were found in the MRI images. These were divided into two tendinopathies of the supraspinatus tendon (9.1%), one labrum change (4.5%), three humeral tuberosity cysts (13.6%), fourteen AC joint hypertrophies (63.6%), five AC joint osteophytes (22.7%), and ten signs of inflammation of the AC joint (45.5%) (see Figure 1).

In more than half of the shoulders examined, namely 63.6%, hypertrophy of the joint could be detected.

## 4. Discussion

The main reason for the study was to find out whether long-term RT causes pathological alterations in the shoulder. In our study of 11 athletes and 22 shoulders, we found that bodybuilding athletes had a low risk of degenerative changes and discomfort in the shoulder. So far, all published studies have been conducted using questionnaires that show which injuries professional weightlifting athletes generally sustain. These show a relatively low risk of injury for bodybuilding, with 0.12–0.70 injuries per year or 0.24 injuries per 1000 h of training [11,12]. Most injuries were reported in the shoulder, elbow, knee, and lumbar spine. Overall, more than 40% of injuries involved the upper extremity [13,14].

In epidemiologic studies, the shoulder complex has been identified as the predominant site of injury in the RT population in general, with prevalence rates ranging from 22% to 36%. A retrospective study of competitive weightlifters from Oceania (n = 101) performed by Keogh et al. [15] looked into injury trends. In their study, the shoulder complex was the most frequently injured site, accounting for 36% of all reported injuries. Goertzen et al. looked at the temporal and point prevalence of injuries in 358 RT patients and found that the shoulder complex was the most prevalent area of injury, accounting for 34% of all injuries [16].

One possible explanation may be that these athletes frequently use large loads and exercises such as the bench press and overhead presses, demonstrated by shoulder injuries making up a significant portion of the weight-training sports injuries identified in the studies. Additionally, the shoulder may be placed in rather unfavorable situations during bench and overhead presses, such as positions involving end-range external rotation while carrying big weights, making the shoulder more vulnerable to acute and chronic injuries [11,16,17].

Weight training puts a lot of strain on the shoulder complex from a biomechanical standpoint since it forces a non-weight-bearing joint to function as a weight-bearing joint while repeatedly lifting heavy weights in adverse positions. The majority of training regimens also place too much emphasis on muscles that grow in size and strength while neglecting essential stabilizing muscles, which may thus impede shoulder function [5,17]. The findings of a study conducted by Barlow et al. [18] on shoulder strength and range of motion (ROM) support this, as they concluded that bodybuilders have imbalances in shoulder strength and ROM that may predispose them to shoulder disorders [18].

During upper extremity RT, the AC joint is subjected to greater stress, predisposing it to a disease known as distal clavicular osteolysis. A widening of the AC joint, subchondral stress fractures, and bone lysis at the distal clavicle where it forms the AC joint are all symptoms of osteolysis of the distal clavicle. As a result of repetitive microtrauma at the AC joint during the lowering phase of the exercise, when the arm is stretched posterior to the trunk, this condition has been linked mainly to the bench press exercise [16].

In their work, Handoll et al. [19] surveyed more than 100 strongman athletes about their injuries in the past years. They found that the shoulder had the second-highest injury rate after the lower back. Traditional strength exercises such as the shoulder press, bench press, and squats place extraordinary stress on the shoulder and frequently result in injury. One of the main reasons for injury cited by more than a quarter of the respondents was incorrect technique in the exercises performed. These studies all refer to the subjective assessments of study participants and cannot offer objective verification.

Barreto et al. [10] studied patients with unilateral shoulder pain using MRI imaging. In addition to the affected side, an MRI of the asymptomatic shoulder was also performed. Abnormalities on this side were also found in imaging results with almost the same frequency. It can be concluded that shoulder abnormalities are present in many individuals, even in the absence of symptoms, despite them not yet having clinical relevance.

In a study by Baretto et al. [10], it was suggested that MRI diagnostics should not be performed without clinical evidence because they have no significance in terms of the current health status of a joint. However, whether these changes cause problems with advancing age was unfortunately not further investigated in this study.

In our study, 22 shoulders were prospectively examined. Demographic information about the subjects and information about their training history was collected through questionnaires. The high training load led to few tendinopathies, but mostly changes in the AC joint were detected.

Among the major strengths of our study is its homogeneous participant cohort. Nevertheless, the study conducted has some limitations. The number of shoulders examined was relatively low at 22. We did not include a suitable control group for comparison purposes. To resolve these limitations, one would have to include a comparison group with asymptomatic individuals in the same age group but without excessive RT. Additionally, a larger group of bodybuilders should be examined. However, the suitable population is not unending, which means that a bigger study probably has to be conducted using a multi-center design.

## 5. Conclusions

According to our study, bodybuilding leads to increased AC joint pathologies in asymptomatic athletes. Other shoulder structures do not seem to be affected. This study contributes to the investigation of further hidden changes that may correlate with RT.

## Figures and Tables

**Figure 1 sports-10-00210-f001:**
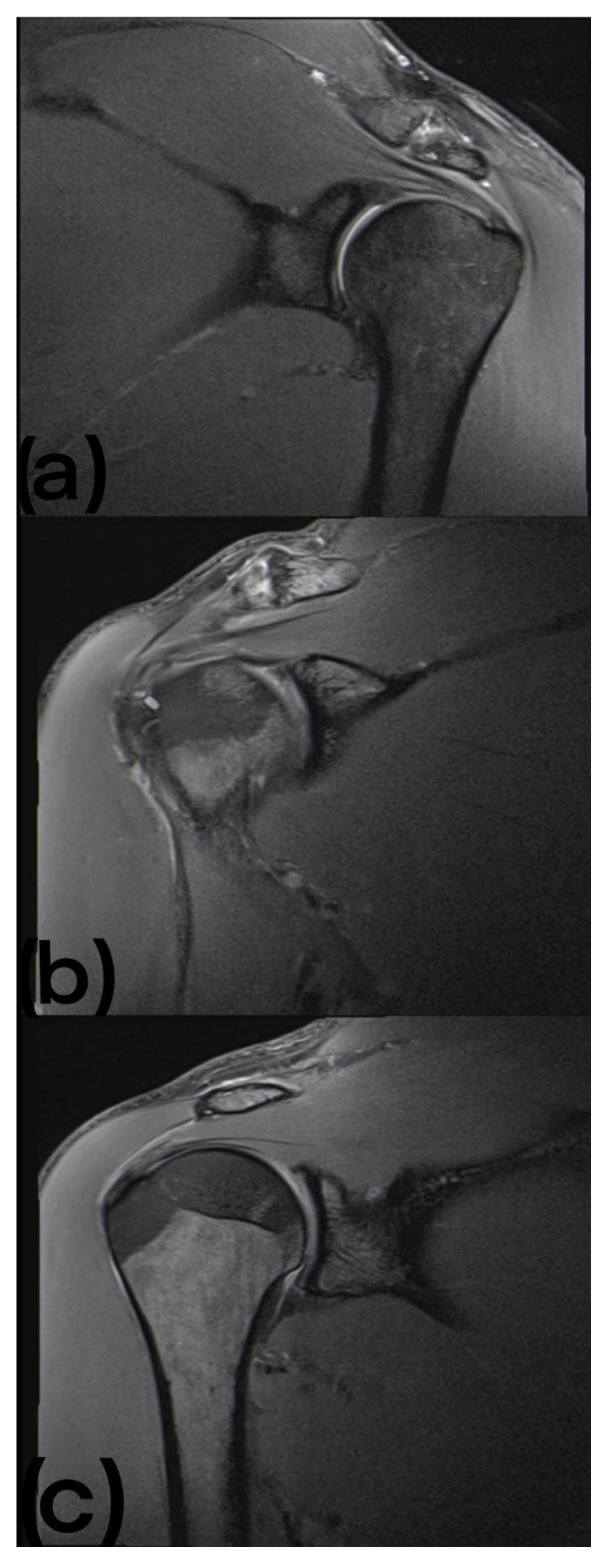
(**a**) MRI of the left shoulder showing edematous changes at the AC joint. (**b**) MRI of the right shoulder showing small cystic resorptions near the intertubercular sulcus. (**c**) MRI of the right shoulder of the same participant shown in (**b**), showing low attachment tendinopathy of the supraspinatus tendon.

**Table 1 sports-10-00210-t001:** Participant demographics.

Participant	Age on the Day of Examination in Years	Gender	Height in cm	Weight in kg	Handiness	BMI in kg/m^2^
1	26	male	180	82	right	25.3
2	25	male	176	85	right	27.4
3	24	male	172	77	right	26
4	28	male	170	78	right	27
5	32	male	179	74	right	23.1
6	29	male	179	94	left	29.3
7	40	male	183	104	right	31.1
8	35	male	175	100	right	32.7
9	41	male	181	96	right	29.3
10	48	male	185	94	right	27.5
11	30	male	176	100	left	32.3
Total mean	32.5	male	177.8	89.5	right	28.3

**Table 2 sports-10-00210-t002:** Checklist of pathologic MRI findings for each shoulder.

MRI Abnormality	Number of Findings, *n* = 22		*n* in %
Tendinopathy			
	Supraspinatus	2	9.1
	Infraspinatus	0	0.0
	Subscapularis	0	0.0
Partial rupture of rotator cuff tendon			
	Supraspinatus	0	0.0
	Infraspinatus	0	0.0
	Subscapularis	0	0.0
Complete rupture of rotator cuff tendon			
	Supraspinatus	0	0.0
	Infraspinatus	0	0.0
	Subscapularis	0	0.0
Acromioclavicular joint			
	Hypertrophy	14	63.6
	Osteophytes	5	22.7
	Inflammatory signs	10	45.5
Long biceps tendon			
	Tendinopathy	0	0.0
	Partial rupture	0	0.0
	Complete rupture	0	0.0
Labrum changes		1	4.5
Fatty infiltration of rotator cuff		0	0.0
Humeral tuberosity cysts		3	13.6
Glenohumeral osteoarthritis		0	0.0

**Table 3 sports-10-00210-t003:** Pain in bodybuilding athletes—Visual Analogue Scale results ^a^.

Subject	Visual Analog Scale			
	Shoulder, during Sports	Other Joints, during Sports	Shoulder, after Sports	Anxiety and Instability
01-right	0	0	1	0
01-left	0	0	1	0
02-right	3	0	4	4
02-left	3	0	4	4
03-right	0	0	0	0
03-left	0	0	0	0
04-right	0	4	0	0
04-left	0	0	0	0
05-right	1	3	1	0
05-left	0	0	0	0
06-right	0	3	0	0
06-left	0	0	0	0
07-right	3	0	3	0
07-left	2	2	3	0
08-right	0	1	0	0
08-left	0	1	0	0
09-right	0	0	0	0
09-left	0	0	0	0
10-right	0	0	0	0
10-left	0	0	0	0
11-right	0	0	0	0
11-left	0	0	0	0
Total mean	0.5	0.6	0.8	0.4

^a^ Shoulder, during sports: 0–10 points. Other joints, during sports: 0–10 points. Shoulder, after sports: 0–10 points. Anxiety and instability: 0–10 points.

**Table 4 sports-10-00210-t004:** Overview showing the Constant Murley Score and its subscores for each shoulder ^a^.

Subject	Pain	Activity	Mobility	Strength	Total
01-right	15	20	40	25	100
01-left	15	20	40	25	100
02-right	10	20	40	25	95
02-left	5	18	40	25	88
03-right	10	20	40	25	95
03-left	10	20	40	25	95
04-right	15	20	40	25	100
04-left	15	20	40	25	100
05-right	15	20	40	25	100
05-left	15	20	40	25	100
06-right	15	20	40	25	100
06-left	15	20	40	25	100
07-right	10	19	40	20	89
07-left	10	19	40	18	87
08-right	15	20	40	25	100
08-left	15	20	40	25	100
09-right	15	20	40	25	100
09-left	15	20	40	25	100
10-right	15	20	40	25	100
10-left	15	20	40	25	100
11-right	15	20	40	25	100
11-left	15	20	40	25	100
Total mean	13.4	19.8	40	24.5	97.7

^a^ Constant Murley Score: 0–100 points. Pain: 0–15 points. Activity: 0–20 points. Mobility: 0–40 points. Strength: 0–25 points.

## Data Availability

Data available on request due to restrictions, e.g., privacy or ethics.

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
