# Peer review of "The Effect of Strength Training on Undetected Shoulder Pathology in Asymptomatic Athletes: An MRI Observational Study"

_sports, 2022, doi:10.3390/sports10120210_

Round 1

Reviewer 1 Report

Line 25: The article does not show that strength training exerts force on the AC joint.  It shows that strength training is associated with MRI documented AC pathology

Line 27: I do not think that 11 patients is enough for a declarative statement that it "does not".  Should be less strong of a statement like "may not".

Line 31-33 is confusing.  Simplify this statement.  i.e. The shoulder has a unique anatomical structure that may make it prone to both acute and chronic injuries

Line 42: is there a reference for injuries from strength training?

Materials and methods section is concise and appropriate.  Any reason why study was limited to 11 athletes?  Was there a preplanned number? 

Results: There should be a demographic and age breakdown of the 11 participants.  Age range, gender, etc.  Also height, weight, BMI, body fat? muscle mass? or other characteristics that will help readers decide on the generalizeability of your fiondings.

Line 111 should be eliminated.  Results section should list the facts.  Authors should leave commentary for the discussion section

Line 122-23: should add "in our study of 11 athletes and 22 shoulders" as this is not a large comprehensive study to ensure all body builders are at low risk

Conclusion is concise and appropriate

Author Response

Dear Reviewer,

Thank you for your patience and valuable comments. 

R: Line 25: The article does not show that strength training exerts force on the AC joint. It shows that strength training is associated with MRI documented AC pathology

A: Thank you very much for your feedback. Yes, you are right. Our study shows that strength training is associated with MRI documented AC pathology. It does not show that strength training exerts force on the AC joint. We have now modified the sentence (line 25).

R: Line 27: I do not think that 11 patients is enough for a declarative statement that it "does not". Should be less strong of a statement like "may not". 

A: Thank you for your advice, we changed the statement according to your remarks (line 26).

R: Line 31-33 is confusing. Simplify this statement. i.e. The shoulder has a unique anatomical structure that may make it prone to both acute and chronic injuries

A: Thank you for the note. We have now simplified the statement (lines 31-33).

R: Line 42: is there a reference for injuries from strength training? 

A: A reference is the study by Kerr, Z.Y.; Collins, C.L.; Comstock, R.D. Epidemiology of weight training-related injuries presenting to United States emergency departments, 1990 to 2007. The American journal of sports medicine 2010, 38, 765-771, doi:10.1177/0363546509351560 or the study by Bonilla: Bonilla, D.A.; Cardozo, L.A.; Vélez-Gutiérrez, J.M.; Arévalo-Rodríguez, A.; Vargas-Molina, S.; Stout, J.R.; Kreider, R.B.; Petro, J.L. Exercise Selection and Common Injuries in Fitness Centers: A Systematic Integrative Review and Practical Recommendations. International journal of environmental research and public health 2022, 19, doi:10.3390/ijerph191912710.

In our study, there were no injuries from strength training. Actually, we have excluded bodybuilders with past injuries. More precisely, the criteria that led to exclusion from our project were a pre-operated shoulder or a previously diagnosed shoulder pathology detected in the last 5 years. We have added a sentence (lines 53-54) in the manuscript to clarify it.

R: Materials and methods section is concise and appropriate. Any reason why study was limited to 11 athletes? Was there a preplanned number? 

A: Thank you very much. We contacted more participants, but got 11 participants with our inclusion criteria. There was no preplanned number. 

R: Results: There should be a demographic and age breakdown of the 11 participants. Age range, gender, etc. Also height, weight, BMI, body fat? muscle mass? or other characteristics that will help readers decide on the generalizeability of your fiondings.

A: Thanks for this advice. We have now included a demographic breakdown in the form of a table in the manuscript (Table 1.).

R: Line 111 should be eliminated. Results section should list the facts. Authors should leave commentary for the discussion section

A: Line 111 ist now eliminated. Thanks for your feedback.

R: Line 122-23: should add "in our study of 11 athletes and 22 shoulders" as this is not a large comprehensive study to ensure all body builders are at low risk

A: Thanks for this remark! We have now added "in our study of 11 athletes and 22 shoulders" (line 128).

R: Conclusion is concise and appropriate

A: Thank you very much.

Reviewer 2 Report

-Dear Author, Thank you for your effort and time for your manuscript. I have provided the necessary recommendations for your paper.

-It is important to congratulate the authors for the work carried out, despite the limitations of the research.

-I propose that more emphasis be given to the importance of the study and the contribution it brings to current knowledge. Also please mention your hypothesis in this section.

-The introduction did not identify the gap in the literature. Basically, the question is why are the authors studying these associations?

-In line 38-40 pelase put the references, the references need to be cited. I think it is not clear that the references 1-4 is enough for a whole paragrapgh.

In lıne 38, what do you mean by high intensity strength training? Do you mean >70%1RM? Please consider and use the best reference for your sentences. I would remommend to read that paper ‘’ Resistance Training Recommendations to Maximize Muscle Hypertrophy in an Athletic Population: Position Stand of the IUSCA’’ DOI: https://doi.org/10.47206/ijsc.v1i1.81

It would be fine to identify the meaning and the differences of ‘’strength training’’, ‘’weight training’’ and ‘’resistance training’’. I suggest using one term for your study. 

Please also present the descriptive statistics of the participants (age, height, weight, BMI, training experience….)

I think it is crucial to put the power of analysis using G-power. Pleae consider to provide sampling power. The authors have already mentioned that the sampling power is  the limitations of the study.

The discussion provides sufficient literature.

In conclusion part, Ppactical implications need to be written in more detail. What is the novelty of your findings?

Author Response

Dear Reviewer,

Thank you for your patience and valuable comments. 

R: Dear Author, Thank you for your effort and time for your manuscript. I have provided the necessary recommendations for your paper. It is important to congratulate the authors for the work carried out, despite the limitations of the research.

A: Thank you very much. We appreciate it much.

R: I propose that more emphasis be given to the importance of the study and the contribution it brings to current knowledge. Also please mention your hypothesis in this section.

A: Thanks for your remarks. We added a sentence about the importance of this study (see lines 48-51). Our hypotheses were that the shoulders would present a high prevalence of alterations and an agreement of the shoulder surgeon with the radiologist regarding the pathological changes.

R: The introduction did not identify the gap in the literature. Basically, the question is why are the authors studying these associations?

A: Thank you for pointing this out. We wanted to know if resistance training has a negative effect to the shoulder morphology and whether it is associated with hidden asymptomatic shoulder pathologies (lines 51-53).

R: In line 38-40 pelase put the references, the references need to be cited. I think it is not clear that the references 1-4 is enough for a whole paragrapgh.

A: We thank you for this comment. We have changed it all according to your remarks.

R: In lıne 38, what do you mean by high intensity strength training? Do you mean >70% 1RM? Please consider and use the best reference for your sentences. I would remommend to read that paper ‘’ Resistance Training Recommendations to Maximize Muscle Hypertrophy in an Athletic Population: Position Stand of the IUSCA’’ DOI: https://doi.org/10.47206/ijsc.v1i1.81

A: Thank you for your recommendation. We have changed and chosen to write resistance training in the whole manuscript.

R: It would be fine to identify the meaning and the differences of ‘’strength training’’, ‘’weight training’’ and ‘’resistance training’’. I suggest using one term for your study.

A: Yes, you are right. We have changed our term to resistance training.

R: Please also present the descriptive statistics of the participants (age, height, weight, BMI, training experience....)

A: Thanks for this advice. We have now included a demographic breakdown in the form of a table in the manuscript (Table 1).

R: I think it is crucial to put the power of analysis using G-power. Pleae consider to provide sampling power. The authors have already mentioned that the sampling power is the limitations of the study.

A:  Thanks for your advice. The actual power is 0.82 and the total sample size is 21. Calculated with G-power.

R: The discussion provides sufficient literature.

A: Thank you very much.

R: In conclusion part, Practical implications need to be written in more detail. What is the novelty of your findings?

A: Yes, you are right. Our study shows as first the outcome of resistance training to the shoulder in asymptomatic patients as shown radiologically. Our study is the first to demonstrate that AC pathology can be detected on imaging in asymptomatic strength athletes. We have added lines 194 - 195.

Reviewer 3 Report

This study aimed to determine whether long-term excessive strength training has a structural effect on the shoulder. These are my comments and suggestions:

Abstract:

Abstract is nicely written with clearly stated the purpose of the study.

Introduction:

In the Introduction section more relevant references should be included that will support the rational of the study.

Line 34: Please add the relevant reference

Line 35-36: Please add the relevant reference

Line 42: Please add the relevant reference to that.

Line 43-45: Please add the relevant literature.

Line 48-49: The references 1-4 are for the aim of the study?  Also, the aim of the study here is different than the abstract.

Materials and Methods:

- The authors mention that the mean age of the participants was 32.5 years (ranged 24-48). The participant(s) aged 48 years started the training 4 years ago?

There is a very wide variation in the age of the participants. Report the standard deviation.

- Line 58: You mention the participants as patients. Why?

- Line 62-65: "The second ......... [5-7]" Was any clinical examination performed by a specialist doctor?

- In Table 1 "Checklist of pathologic MRI findings for each shoulder" there is a misunderstanding. It is mentioned that in Tendinopathy of Supraspinatus muscle, (n in %) was 9.1%. Was it equally distributed to both supraspinatus muscles? The same question goes to the other finding of the table. 

Line 115-118: The MRI in Figure 1 is from the same participant? What is the age and the years of training of that participant?

General comments:

You should mention the years of training prior to the examination.

Did you consider any possible injuries prior to the 4 years of training?

You should provide more information about participants regarding their participation in professional bodybuilding. In what age and after how many years of training?

The injuries (abnormalities) after how many years of training were reported?

The limitation section provide sufficient information about the outcomes of the study.

Author Response

Dear Reviewer,

Thank you for your patience and valuable comments. 

R: This study aimed to determine whether long-term excessive strength training has a structural effect on the shoulder. These are my comments and suggestions:

A: Thank you for your review.

R: Abstract:

Abstract is nicely written with clearly stated the purpose of the study.

A: Thank you very much.

R: Introduction:

In the Introduction section more relevant references should be included that will support the rational of the study.

A: Yes, you are right. We have now provided more relevant references on it (publication year 2022).

R: Line 34: Please add the relevant reference

Line 35-36: Please add the relevant reference

Line 42: Please add the relevant reference to that.

Line 43-45: Please add the relevant literature.

A: Thank you for the comments. All the relevant references are now added.

R: Line 48-49: The references 1-4 are for the aim of the study? Also, the aim of the study here is different than the abstract.

A: Thank you for the note. Yes, the references 1-4 are for the aim of the study. I have corrected it. We have modified the sentence „…performed several times a week for years leads to damage to the glenohumeral joint and surrounding structures.“ to  „… performed several times a week for years leads to hidden changes to the glenohumeral joint and surrounding structures.“ and added „The purpose of this study was to determine if persistent, long-term RT in the shoulder results in undetected pathological changes.“ (lines 45-46)

R: Materials and Methods:

- The authors mention that the mean age of the participants was 32.5 years (ranged 24-48). The participant(s) aged 48 years started the training 4 years ago?

A: The participant aged 48 years trained over 10 years.

R: There is a very wide variation in the age of the participants. Report the standard deviation.

A: Thanks for the remark. The standard deviation for the age is 7.3. (see line 60)

R: - Line 58: You mention the participants as patients. Why?

A: Thank you for pointing this out. We have now changed the word patient to participant (line 59).

R: - Line 62-65: "The second ......... [5-7]" Was any clinical examination performed by a specialist doctor?

A: An orthopedic resident performed the clinical examinations. 

R: - In Table 1 "Checklist of pathologic MRI findings for each shoulder" there is a misunderstanding. It is mentioned that in Tendinopathy of Supraspinatus muscle, (n in %) was 9.1%. Was it equally distributed to both supraspinatus muscles? The same question goes to the other finding of the table.

A: We examined 22 shoulders and in 2 (9.1%) of them a Supraspinatus muscle tendinopathy was found. The same goes to the other findings of the table. We regarded each Supraspinatus muscle as a separate sample.

R: Line 115-118: The MRI in Figure 1 is from the same participant? What is the age and the years of training of that participant?

A: The MRI in Figure 1 is from 2 different patients. The above image a) shows the left shoulder (also dominant side) of patient 11 from Table 1 and the other images, namely b) and c), show the right shoulder (also dominant side) of patient 9 from Table 1. The data of the patients can be derived from Table 1.

R: General comments:

You should mention the years of training prior to the examination.

A: Thank you for your note. All participants had at least 4 years of training prior to the examination (see line 62).

R: Did you consider any possible injuries prior to the 4 years of training?

A: Yes, we excluded all participants who had injuries and pathologies in the past history. We have written a sentence to clarify this (see lines 57-58).

R: You should provide more information about participants regarding their participation in professional bodybuilding. In what age and after how many years of training?

A: Seven participants older than the mean age of 32.5 years trained more than 9 years (see lines 63-64).

R: The injuries (abnormalities) after how many years of training were reported?

A: After at least 4 years of training (see lines 55-56).

R: The limitation section provide sufficient information about the outcomes of the study.

A: Thank you for this comment.

Round 2

Reviewer 3 Report

I am satisfied with the response, improvements and corrections of the manuscript